# Targeted PARP Inhibition Combined with FGFR1 Blockade is Synthetically Lethal to Malignant Cells in Patients with Pancreatic Cancer

**DOI:** 10.3390/cells9040911

**Published:** 2020-04-08

**Authors:** Shiue-Wei Lai, Oluwaseun Adebayo Bamodu, Jia-Hong Chen, Alexander TH Wu, Wei-Hwa Lee, Tsu-Yi Chao, Chi-Tai Yeh

**Affiliations:** 1Graduate Institute of Clinical Medicine, Taipei Medical University, Taipei 110, Taiwan; xsurfer@office365.ndmctsgh.edu.tw (S.-W.L.); ndmc_tw.tw@yahoo.com.tw (J.-H.C.); 2Division of Hematology-Oncology, Department of Internal Medicine, Tri-Service General Hospital, National Defense Medical Center, Taipei 114, Taiwan; 3Department of Internal Medicine, Tri-Service General Hospital Penghu Branch, Penghu 880, Taiwan; 4Department of Hematology and Oncology, Cancer Center, Taipei Medical University-Shuang Ho Hospital, New Taipei City 235, Taiwan; 16625@s.tmu.edu.tw; 5Department of Medical Research & Education, Taipei Medical University-Shuang Ho Hospital, New Taipei City 235, Taiwan; whlpath97616@s.tmu.edu.tw; 6The PhD Program for Translational Medicine, College of Medical Science and Technology, Taipei Medical University and Academia Sinica, Taipei 110, Taiwan; chaw1211@tmu.edu.tw; 7Graduate Institute of Medical Sciences, National Defense Medical Center, Taipei 114, Taiwan; 8Department of Pathology, Taipei Medical University-Shuang Ho Hospital, New Taipei City 235, Taiwan; 9Taipei Cancer Center, Taipei Medical University, Taipei City 110, Taiwan; 10Department of Medical Laboratory Science and Biotechnology, Yuanpei University of Medical Technology, Hsinchu City 30015, Taiwan

**Keywords:** pancreatic cancer, PDAC, FGFR1, PD173074, PARP, olaparib, dasatinib, selective inhibitor, FGFR1 inhibitor-resistance, FGFR1/PARP signaling

## Abstract

The role and therapeutic promise of poly-ADP ribose polymerase (PARP) inhibitors in anticancer chemotherapy are increasingly being explored, particularly in adjuvant or maintenance therapy, considering their low efficacy as monotherapy agents and their potentiating effects on concurrently administered contemporary chemotherapeutics. Against the background of increasing acquired resistance to FGFR1 inhibitors and our previous work, which partially demonstrated the caspase-3/PARP-mediated antitumor and antimetastatic efficacy of PD173074, a selective FGFR1 inhibitor, against ALDH-high/FGFR1-rich pancreatic ductal adenocarcinoma (PDAC) cells, we investigated the probable synthetic lethality and therapeutic efficacy of targeted PARP inhibition combined with FGFR1 blockade in patients with PDAC. Using bioinformatics-based analyses of gene expression profiles, co-occurrence and mutual exclusivity, molecular docking, immunofluorescence staining, clonogenicity, Western blotting, cell viability or cytotoxicity screening, and tumorsphere formation assays, we demonstrated that FGFR1 and PARP co-occur, form a complex, and reduce survival in patients with PDAC. Furthermore, FGFR1 and PARP expression was upregulated in FGFR1 inhibitor (dasatinib)-resistant PDAC cell lines SU8686, MiaPaCa2, and PANC-1 compared with that in sensitive cell lines Panc0403, Panc0504, Panc1005, and SUIT-2. Compared with the limited effect of single-agent olaparib (PARP inhibitor) or PD173074 on PANC-1 and SUIT-2 cells, low-dose combination (olaparib + PD173074) treatment significantly, dose-dependently, and synergistically reduced cell viability, upregulated cleaved PARP, pro-caspase (CASP)-9, cleaved-CASP9, and cleaved-CASP3 protein expression, and downregulated Bcl-xL protein expression. Furthermore, combination treatment markedly suppressed the clonogenicity and tumorsphere formation efficiency of PDAC cells regardless of FGFR1 inhibitor-resistance status and enhanced RAD51 and γ-H2AX immunoreactivity. In vivo studies have shown that both early and late initiation of combination therapy markedly suppressed tumor xenograft growth and increase in weight, although the effect was more pronounced in the early initiation group. In conclusion, FGFR1 inhibitor-resistant PDAC cells exhibited sensitivity to PD173074 after olaparib-mediated loss of PARP signaling. The present FGFR1/PARP-mediated synthetic lethality proof-of-concept study provided preclinical evidence of the feasibility and therapeutic efficacy of combinatorial FGFR1/PARP1 inhibition in human PDAC cell lines.

## 1. Introduction

Pancreatic cancer is the seventh leading cause of cancer-related deaths globally; it accounts for more than 20% of all deaths caused by abdominal cancers and approximately 5% of all cancer-related deaths [1]. Most deaths are histologically attributed to the pancreatic ductal adenocarcinoma (PDAC) subtype, which originates from the pancreatic duct epithelium, represents approximately 85% of the total incidence of pancreatic cancer, and is often unresectable; consequently, patients have limited treatment options and exhibit high morbidity and mortality rates [1,2]. A yearly increase was reported in PDAC-specific mortality rates (approximately 98% mortality/incidence), and a consistently low five-year survival rate of <5% was reported over the past decade [1]. 

Currently, the standard of care is gemcitabine with nanoparticle albumin-bound paclitaxel, erlotinib, or capecitabine, while for advanced stage (III and IV) or metastatic PDAC, the preferred chemotherapy is a combination of fluorouracil, leucovorin, irinotecan, and oxaliplatin, known as FOLFIRINOX chemotherapy; however, these chemotherapeutic drugs are associated with a high risk of severe drug-related toxicity, acquired resistance, and non-significant survival benefits [3,4]. Hence, the discovery of new actionable molecular oncotargets is necessary. Furthermore, screening methods should be developed for identifying novel and highly efficacious therapeutic agents that inhibit disease progression, alleviate resistance to treatment, and improve prognosis in patients with PDAC. 

The involvement of highly conserved and ubiquitous transmembrane tyrosine kinase receptors, namely fibroblast growth factor receptors (FGFRs), in tumorigenesis, cell fate determination, survival, motility, angiogenesis, and malignantization of tumor cells, as well as in reduced sensitivity to anticancer therapeutics and poor prognosis is extensively documented [5]. Notably, genetic alterations in FGFR1, such as amplification or an increase in gene copy number, are positively correlated with overexpression and are more prevalent than genetic aberrations in FGFR2, FGFR3, and FGFR4 [5,6]. Hence, Lehnen et al., studying the role of the FGFR1 gene copy number and expression pattern in patients with PDAC, suggested that the association between FGFR1 amplification, mRNA or protein expression, and the proliferative potential of PDAC cells can be exploited for therapeutic purposes using FGFR1 inhibitors [7]. This is consistent with results indicating that enhanced FGFR signaling predisposed pancreatic cancer cells to the killing effect of dovitinib, a potent FGFR inhibitor, in preclinical models [8], thus indicating that the pharmacological or molecular targeting of FGFR1 in patients with PDAC has therapeutic potential. However, despite the actual or predicted therapeutic benefits of FGFR1-targeting therapeutics, the post-treatment development of resistance due to accrue de novo genetic alterations in the kinase targets of these therapeutics following extended exposure to FGFR1 inhibitors is extensively documented [9,10,11]. Therefore, the discovery or development of adjuvant or maintenance therapeutic agents for treating patients with PDAC harboring FGFR1 aberrations is necessary.

Poly (ADP ribose) polymerase (PARP) plays a critical role in the sensing of genomic damage signals, DNA repair and replication, inflammation, transcriptional and posttranscriptional gene expression modulation, and cell death regulation. PARP has also been implicated in several pathological processes, including carcinogenesis, by PARylation, direct or mediated interactions with oncogenes, and regulation of oncogenic transcription factors [12]; thus the increased interest in and exploration of the therapeutic potential and clinical feasibility of PARP inhibition as an effective anticancer strategy in the last decade. Several PARP inhibitors are being developed or are currently undergoing clinical evaluation. As the concept of synthetic lethality in anticancer therapy continues to gain traction, the probable exploitability of the DNA repair-disrupting potential of PARP inhibitors (PARPi) is receiving a second look. In the context of the present study, synthetic lethality is implied if the impairment of either of two oncogenic pathways is not lethal or sufficiently detrimental to the viability of cancerous cells, but their combination induces lethality in the cancerous cells. An increasing amount of preclinical and clinical evidence suggests that PARPi may potentiate the effect of conventional chemotherapeutics, enhance sensitivity to radiation therapy, and serve as adjuvants or maintenance therapeutic agents [12].

Against the background of the increasing incidence of acquired resistance to the FGFR1 inhibitor and our previous work that partially demonstrated the caspase-3/PARP-mediated anticancer and anti-metastasis efficacy of PD173074, a selective FGFR1 inhibitor, against aldehyde dehydrogenase (ALDH)-rich or FGFR1-high PDAC cells [13], the present study investigates the probable synthetic lethality and therapeutic efficacy of targeted combinatorial FGFR1/PARP inhibition in patients with PDAC harboring FGFR1 aberrations using a multifaceted approach, including bioinformatics-aided prediction and visualization, drug screening, molecular biology, and functional assays.

## 2. Materials and Methods

### 2.1. Analysis of Public Cancer Datasets

Public online cancer databases, namely The Cancer Genome Atlas (TCGA), Gene Expression Omnibus (GEO), and Broad Institute Cancer Cell Line Encyclopedia, were used in this study. We probed the provisional TCGA pancreatic adenocarcinoma (TCGA PAAD) Illumina HiSeq RNAseq dataset (*n* = 186) through the University of California Santa Cruz Cancer Browser (https://xenabrowser.net/heatmap/) and the GEO Illumina Human HT-12 V4.0 Expression BeadChip GSE59357/GPL10558/GDS5627 dataset on the gene expression profile in pancreatic carcinoma cell lines that are resistant or sensitive to dasatinib, a U.S. FDA-approved small-molecule kinase inhibitor for the treatment of pancreatic cancer (https://www.ncbi.nlm.nih.gov/sites/GDSbrowser?acc=GDS5627). We also used the AFFY_HG_U133_PLUS_2 dataset GSE17891/GPL570, which originally investigated the pervasive subtypes of PDAC and their different responses to anticancer treatment (*n* = 47 samples, 54,675 genes) (https://www.ncbi.nlm.nih.gov/geo/geo2r/?acc=GSE17891&platform=GPL570). 

### 2.2. Drugs and Reagents

PD173074 (Sigma-P2499, HPLC ≥ 96%) and olaparib (AZD2281/KU0059436, #S1060, HPLC ≥ 99.7%) were purchased from Sigma-Aldrich Co. (St. Louis, MO, USA) and Selleck Chemicals (Antibody International Inc. Jhubei City, Hsinchu County, Taiwan), respectively. Stock solutions (1 mM) of each drug were prepared by dissolution in phosphate-buffered saline (PBS) and stored in a dark room at −20 °C. PBS, dimethyl sulfoxide (DMSO), sulforhodamine B (SRB) reagent, trypsin/ethylenediaminetetraacetic acid, Tris aminomethane (Tris) base, and acetic acid were purchased from Sigma-Aldrich Co. (St. Louis, MO, USA). Dulbecco’s modified Eagle’s medium (DMEM) was purchased from Invitrogen (Invitrogen Life Technologies, Carlsbad, CA, USA). 

### 2.3. Cell lines and Culture 

Human PDAC cell lines PANC-1 (ATCC^®^ CRL-1469), AsPC-1 (ATCC^®^ CRL-1682), and PANC 0403 (ATCC^®^ CRL-2555) were obtained from American Type Culture Collection (ATCC Manassas, VA, USA), and SUIT-2 (Japanese Collection of Research Bioresources Cell Bank [JCRB]1094) cells were obtained from the National Institute of Biomedical Innovation, Health and Nutrition (JCRB Cell Bank, Japan). The PANC-1 and SUIT-2 cells were cultured in DMEM (Invitrogen Life Technologies, Carlsbad, CA, USA). Culture media were supplemented with 10% fetal bovine serum and 1% penicillin–streptomycin (Invitrogen, Life Technologies, Carlsbad, CA, USA). The cells were incubated in a 5% humidified CO_2_ incubator at 37 °C. The cells were subcultured at 100% confluence every 48–72 h. The vendors identified and authenticated the cell lines on the basis of karyotype and short tandem repeat analyses, and our team regularly checked the cells to confirm that they were free from mycoplasma contamination. The PDAC cells were treated with indicated concentrations of olaparib and/or PD173074. 

### 2.4. Sulforhodamine B Cytotoxicity Assay

The PANC-1 or SUIT-2 cells were seeded at a density of 3 × 10^3^ cells/well in 96 well plates in triplicate and were cultivated for 24 h. Then, the cells were treated with olaparib and/or PD173074 for 48 h, fixed with 10% trichloroacetic acid, washed carefully with double-distilled water, and stained using a 0.4% 0.4: 1 (*w/v*) SRB/acetic acid solution. After the unbound dye was removed by carefully washing three times with 1% acetic acid before air-drying the plates, the bound SRB dye was solubilized in 10 mM Tris base, and absorbance was read at 570 nm by using a Molecular Devices Spectramax M3 multimode microplate reader (Molecular Devices LLC., San Jose, CA, USA). The absorbance was positively correlated with cell number over a wide range of values.

### 2.5. Western Blot Analysis

Protein samples (20 μg) were separated using a 10% sodium dodecyl sulfate-polyacrylamide electrophoresis gel and transferred to polyvinylidene fluoride (PVDF) membranes by using a Bio-Rad Mini-Protein electro-transfer system (Bio-Rad Laboratories, Inc., CA, USA). Next, the PVDF membranes were blocked in 5% nonfat milk in Tris-buffered saline with Tween 20 for 1 h. After blocking, the membranes were incubated at 4 °C overnight with primary antibodies against FGFR1 (#9740; 1:1000, Cell Signaling Technology Inc., Danvers, MA, USA), PARP (#9532; 1:1000, Cell Signaling Technology), cleaved PARP (#5625; 1:1000, Cell Signaling Technology), caspase-9 (#9508; 1:1000, Cell Signaling Technology), caspase-3 (#9668; 1:1000, Cell Signaling Technology), Bcl-xL (#2764; 1:1000, Cell Signaling Technology), and β-actin (sc-69879; 1:500, Santa Cruz Biotechnology, Santa Cruz, CA, USA). All the antibodies used are listed in Appendix A. The membranes were then incubated with appropriate horseradish peroxidase (HRP)-conjugated secondary antibodies at room temperature for 1 h and washed carefully with PBS three times. Next, protein band detection was performed using the enhanced chemiluminescence detection system (Thermo Fisher Scientific Inc., Waltham, MA, USA). The protein bands were quantified using ImageJ (https://imagej.nih.gov/ij/). 

### 2.6. Tumorsphere Formation Assay

The PANC-1 or SUIT-2 cells pretreated with indicated concentrations of olaparib and/or PD173074 for 24 h were seeded at a density of 5 × 10^4^ cells/well in 6 well nonadherent plates (Corning Inc., Corning, NY, USA) in DMEM supplemented with B27 supplement (Invitrogen, Carlsbad, CA, USA), 20 ng/mL basic fibroblast growth factor (Invitrogen, Carlsbad, CA, USA), and 20 ng/mL epidermal growth factor (Millipore, Bedford, MA, USA). The cells were cultured for 2 weeks, and tumorspheres formed were counted using an inverted phase-contrast microscope.

### 2.7. Colony Formation Assay

The colony formation capability of the cells was evaluated as previously described [14]. Briefly, the PANC-1 or SUIT-2 cells pretreated with indicated concentrations of olaparib and/or PD173074 for 24 h were seeded at a density of 1 × 10^3^ cells/well in 6 well plates and cultured in a 5% humidified CO_2_ incubator at 37 °C for 15 days. Thereafter, the colonies formed (>50 cells/colony) were stained with crystal violet dye, photographed, and counted.

### 2.8. Immunofluorescence Staining

The PANC-1 or SUIT-2 cells were plated at a density of 2 × 10^4^ cells/well in 6 well chamber slides (Nunc™, Thermo Fisher Scientific) for 24 h. For γ-H2AX staining or RAD51 foci quantification to assess DNA damage repair, the seeded cells were incubated with indicated concentrations of olaparib and/or PD173074 for 48 h and fixed with 4% paraformaldehyde at room temperature for 20 min. The cells were then washed with PBS, permeabilized using 0.1% Triton X-100 in 0.01 M PBS (pH 7.4) for 5 min, blocked in 0.2% bovine serum albumin for 1 h, air dried, and rehydrated in PBS. Next, the cells were incubated with a rabbit polyclonal antibody against histone H2AX (#07-627, Merck KGaA, Darmstadt, Germany) or Rad51 (#07-1782, Merck KGaA, Darmstadt, Germany) diluted to 1:500 in PBS containing 3% normal goat serum at room temperature for 2 h, washed three times in PBS for 10 min each, followed by incubation with an anti-rabbit IgG fluorescein isothiocyanate-conjugated secondary antibody (Jackson ImmunoResearch Inc., West Grove, PA, USA) diluted 1:500 in PBS for 1 h at room temperature. The cells were washed in PBS and mounted using Vectashield mounting medium (Vector Laboratories, Burlingame, CA, USA) with 4′,6-diamidino-2-phenylindole (DAPI) for nuclear staining. Cell imaging was performed using a Zeiss Axiophot fluorescence microscope (Carl Zeiss Microscopy LLC, Thornwood, NY, USA).

### 2.9. Tumor Xenograft Studies

Female BALB/c mice (4–6 weeks old, *n* = 40, median weight = 12.7 ± 2.1 g) were purchased from BioLASCO (BioLASCO Taiwan Co. Ltd., Taipei, Taiwan) and maintained under specific pathogen-free condition with free access to rodent chow and water. The mice were subcutaneously inoculated with 5 × 10^4^ PANC-1 tumorsphere cells suspended in 100 μL of serum-free medium. The mice were randomly divided into two treatment regime groups, namely early start (*n* = 20; treatment started 72 h after inoculation with PANC-1 cells) and late start (*n* = 20; treatment initiated 3 weeks after inoculation with PANC-1 cells) groups. Each group was randomly divided into four treatment subgroups, namely vehicle-treated control, olaparib-treated, PD173074-treated, or combination-treated (olaparib+PD173074) subgroups (*n* = 5 in each subgroup). Randomization was conducted by blindly picking numbers from a dish. The mice were treated with 50 mg/kg olaparib and/or 20 mg/kg PD173074 by intraperitoneal injection daily for 5 consecutive days per week until the last day of Week 9, regardless of the treatment group. The mice in the control group were injected with PBS. Tumor volume (mm^3^) was measured weekly. The formula used for calculating volume was volume = (L × W^2^)/2, where L and W represent the longest and shortest diameters, respectively. The mice were humanely sacrificed at the end of the experimental period, and tumors were resected and weighed. All animal experiments were approved by the Institutional Animal Care and Use Committee of National Defense Medical Center (IACUC-19-101), and all experiments complied with guidelines provided in The National Academies of Science, Engineering and Medicine Guide for the Care and Use of Laboratory Animals [15].

### 2.10. Immunocytochemical Staining

Sections from tumor samples resected from the PD173074 and/or olaparib-treated tumor-bearing BALB/c mice were stained using primary antibodies against FGFR1 (#9740; 1:400, Cell Signaling Technology Inc., Danvers, MA, USA), cleaved PARP (#5625; 1:400, Cell Signaling Technology), Ki-67 (#ab15580; 1:500, Abcam, Cambridge, MA, USA), and Bcl-2 (#15071; 1:400, Cell Signaling Technology). Furthermore, immunohistochemical staining was performed according to the manufacturer’s instructions. Briefly, after washing coverslips with PBS (pH 7.4), they were incubated in 3% H_2_O_2_ for 10 min and then treated with appropriate primary antibodies at 37 °C in a humid chamber for 60 min. Next, the coverslips were stained with the biotinylated HRP-conjugated streptavidin system (#N100, Thermo Fisher Scientific Inc., Bartlesville, OK, USA) before imaging under microscope.

### 2.11. Statistical Analysis

All data represent the mean ± standard deviation of experiments performed at least three independent times in triplicates. A two-sided Student’s *t*-test was used for comparing between two groups, whereas one-way analysis of variance and Tukey’s post hoc test were used for multiple comparisons. Cumulative survival rates were calculated using the Kaplan–Meier method, and the significance of differences in the survival rate was analyzed using the log-rank test. Statistical analyses were performed using GraphPad Prism Version 7 for Windows (GraphPad Software, Inc., San Diego, CA, USA). A *p*-value of <0.05 was considered statistically significant.

## 3. Results

### 3.1. Enhanced FGFR1 and PARP1 Expression Characterizes FGFR1 Inhibitor-Resistant Pancreatic Cells at mRNA and Protein Levels

After our in-house mining of high-throughput gene expression data from public databases to understand the role of FGFR1 and PARP1 in PDAC, we performed computational analyses of the FGFR1 and PARP1 RNAseq expression profile in six PDAC cell lines that were classified according to their sensitivity to the FGFR inhibitor (FGFRi), dasatinib, for evaluating differential expression. Our results indicated that the cumulative median expression of FGFR1 (ILMN_1796229) mRNA was 1.802-fold (*p* < 0.05) higher in the poorly-differentiated dasatinib-resistant pancreatic cancer cells (SU8686, MiaPaCa2, and PANC-1) than in dasatinib-sensitive cells (PANC0403, PANC0504, and PANC1005) (Figure 1A, upper). Similar to FGFR1, PARP1 mRNA expression (ILMN_1686871) was approximately 1.81-fold (*p* < 0.01) higher in dasatinib-resistant pancreatic cancer cells than in dasatinib-sensitive cells (Figure 1A, lower). In parallel with mRNA assays, Western blot results showed that FGFR1 and PARP1 protein expression levels were 1.55- and 1.84-fold higher, respectively, in poorly differentiated FGFRi-resistant primary tumor PANC-1 cells than FGFRi-sensitive metastatic SUIT-2 cells (Figure 1B). We also observed a significant reduction in the viability of PANC-1 cells treated with 1–10 μM PD173074 (2%–71%; IC_50_ = 5.13 μM) or 1–10 μM olaparib/AZD2281 (10%–93%; IC_50_ = 3.70 μM). Similarly, treatment of SUIT-2 cells with 1–10 μM PD173074 or olaparib suppressed their viability by 7%–80% with an IC_50_ of 4.54 μM or 3.62 μM, respectively (Figure 1C). In addition, treatment with 4.5 μM PD173074 or 3.5 μM olaparib significantly downregulated the expression of PARP or FGFR1 protein, respectively, in PANC-1 and SUIT-2 cells (Figure 1D).

### 3.2. Aberrant FGFR1 and PARP1 Expression is Associated with a Cancer Stem Cell-Like Phenotype, Regulates DNA Repair, and Modulates Response to Therapy in PDAC Cells

Enhanced FGFR1 and PARP1 expression characterize FGFR1 inhibitor-resistant pancreatic cells at mRNA and protein levels. Hence, we used the cBioPortal for cancer genomics (https://www.cbioportal.org/) to analyze FGFR1 and PARP1 expression and their functional association with selected markers of cancer stemness, resistance to therapy, and DNA repair by using the provisional TCGA pancreatic adenocarcinoma cohort (*n* = 185 patients/186 samples). Analysis of the TCGA cohort data revealed that patients with high FGFR1 mRNA expression exhibited high mRNA expression of the biomarkers of apoptotic DNA damage (PARP1), cancer stemness (aldehyde dehydrogenase 1, ALDH1A1), and multidrug resistance (ABCB1/MDR1), with concurrent absence of markers of DNA repair (RAD51 and H2AFX) (Figure 2A). We also observed a similar inverse correlation between the highly co-expressed FGFR1, PARP1, ALDH1A1, and ABCB1 mRNAs and the mildly expressed RAD51 and H2AFX mRNAs in the same TCGA cohort (Figure 2B, and Appendix A). Next, multidimensional scaling factor analysis with k-means clustering for gene stratification into clusters based on previously indicated functional similarities between variables/genes showed the nesting of FGFR1, PARP1, ALDH1A1, and ABCB1 into one cluster and H2AFX and RAD51 into another cluster; with the two clusters being diametrically distant (Figure 2C,D). Because of the implication of FGFR1, PARP1, ALDH1A1, ABCB1, RAD51, and H2AFX in therapy response, we further performed gene-set expression-based cohort stratification into therapy responders and non-responders. We performed the principal component analysis of Collisson et al.’s E-GEOD-17891 AFFY_HG_U133_PLUS_2 dataset (*n* = 47) on the pervasive subtypes of PDAC and the difference in their response to therapy [16]. We observed that the *Gene set_high* patients with high FGFR1, PARP1, ALDH1A1, and ABCB1 but low RAD51 and H2AFX levels (*n* = 27) were less sensitive to erlotinib, a receptor tyrosine kinase inhibitor (RTKi), and gemcitabine, a chemotherapeutic nucleoside analog, than the *Gene set_low* patients with low FGFR1, PARP1, ALDH1A1, and ABCB1, but high RAD51 and H2AFX levels (n = 20) (Figure 2E and Appendix A).

### 3.3. FGFR1 and PARP1 Co-Occur and Directly Interact with Each Other at the Expense of RAD51/H2AFX Signaling and Exhibit Adverse Prognostic Implications

To further characterize the functional association between FGFR1, PARP1, ALDH1A1, ABCB1, RAD51, and H2AFX, as well as to provide a therapeutic context, we performed a co-occurrence and mutual exclusivity analysis using the cBioPortal for cancer genomics platform (https://www.cbioportal.org/). Our results indicated that while FGFR1, PARP1, ALDH1A1, and ABCB1 exhibited varying degrees of co-occurrence, their occurrence was mutually exclusive to that of RAD51 or H2AFX (Figure 3A). Reanalysis of the TCGA-PAAD cohort (*n* = 176) showed moderate positive correlation between FGFR1 and PARP1 mRNA expression (Spearman’s R: 0.4; Pearson’s R: 0.1) (Figure 3B). Furthermore, using Schrödinger’s PyMOL molecular graphics system (https://pymol.org/2/) for spatiotemporal visualization of probable molecular interaction between FGFR1 and PARP1, we demonstrated that FGFR1 (Protein Data Bank, PDB: 3C4F) directly binds with PARP1 (PDB: 5DS3) with a complementarity score of 18,008 to form a stable complex with an interface area of 2482.7 Å^2^ and atomic contact energy (ACE) of 286.24 kcal/mol (Figure 3C). In parallel assays, consistent with the in silico data, the results of our co-immunoprecipitation analyses demonstrated direct interaction between FGFR1 and PARP1 in FGFR1-reconstituted PANC-1 and SUIT-2 cells compared with their IgG-treated control counterparts (Figure 3D). By immunofluorescence, we found that PARP1 partially colocalized with FGFR1 in the cell nucleus in the FGFR1-expressing, PARP1-expressing PANC-1 cancer cell line (Figure 3E). In addition, we demonstrated that compared with patients with low PARP1 expression, those with high PARP1 expression exhibited a 1.69-fold worse overall survival (OS) with a mortality index of 0.962 and hazards ratio (HR) = 2.03 (confidence interval, CI: 0.82–5.01) (Figure 3F). Compared to patients with low FGFR1 expression, those with high FGFR1 expression exhibited about 5.25-fold worse OS with a mortality index of 0.974 HR of 8.57 (CI: 2.11–34.91) (Figure 3G).

### 3.4. Olaparib-Induced Pharmacological Inhibition of PARP1 Expression Synergistically Enhances the Therapeutic Effect of FGFR1 Inhibitor PD173074 Through a Caspase-Dependent Apoptotic Mechanism

Having shown that FGFR1 and PARP1 co-occur and directly interact with each other at the expense of RAD51/H2AFX signaling, as well as adversely affect prognosis, we examined whether the FGFR1/PARP1 interaction could be exploited to enhance the anticancer effect of FGFR1 inhibition or overcome resistance to the latter. Our results showed that single-agent treatment with 3.5 μM olaparib or 4.5 μM PD173074 resulted in 47% or 30% loss of PANC-1 cell viability, respectively (*p* < 0.05), and 68% or 52% loss of SUIT-2 cell viability, respectively (*p* < 0.05). However, combination treatment with olaparib/PD173074 induced a 67% (*p* < 0.01) reduction in the viability of PANC-1 cells and an 89% (*p* < 0.01) reduction in SUIT-2 cell viability (Figure 4A). Furthermore, we showed that 0.5–5 μM olaparib dose-dependently enhanced the killing effect of PD173074 in PANC-1 cells, with 1 μM olaparib increasing the PANC-1 cell-killing effect of 2.5, 5, and 10 μM PD173074 by 16%, 13%, and 12%, respectively; while a 39%, 42%, and 44% increase in the cytotoxic effects of 2.5, 5, and 10 μM PD173074, respectively, was observed when combined with 5 μM olaparib (Figure 4B). To determine whether olaparib enhanced the effect of PD173034 through a synergistic or additive mechanism, we performed drug combination analyses by using Chou–Talalay’s algorithm-based isobologram to evaluate dual-agent inhibitory effects and combination indices. As shown by the isobologram, all combination dose-points, except two, were within the right-angled isobologram triangle, indicating olaparib-PD173074 synergism in the PANC-1 cells (Figure 4C). In parallel experiments to elucidate the mechanism underlying the observed anticancer effect of olaparib and/or PD173074, we demonstrated that compared with the mild-to-moderate upregulation of cleaved PARP/PARP, cleaved-caspase (CASP)-9/CASP9, and cleaved-CASP3/CASP3 ratios in PANC-1 cells treated with 4.5 μM PD173074 or 3.5 μM olaparib alone, combining PD173074 with olaparib significantly increased the cleaved PARP/PARP, cleaved- CASP9/CASP9, and cleaved-CASP3/CASP3 ratios, concomitantly with a marked decrease in the expression level of Bcl-xL protein (Figure 4D); thus indicating, at least in part, that the synergistic effect of PD173074 and olaparib is caspase-dependent and mediated by activation of CASP3/CASP9/PARP cell death signaling.

### 3.5. Olaparib Alone or Synergistically with PD173074 Suppresses the Oncogenic Cancer Stem Cell-Like Phenotype of PDAC Cells Through Apoptosis-Related Impairment of DNA Repair

Having shown that the pharmacological inhibition of PARP1 expression synergistically enhances the therapeutic effects of the FGFR1 inhibitor PD173074 through a caspase-dependent apoptotic mechanism, against the background that the cleavage/activation of caspases induces DNA fragmentation, cytoskeletal and nucleosomal degradation, protein cross-linking, the formation of apoptotic bodies, and phagocytosis, with subsequent cell death [17,18], we further sought to understand the effect of this synergism on the cancer stem cell-like phenotype of PDAC cells. We investigated the probable effect of olaparib and/or PD173074 on PDAC stem cells. Our results demonstrated that compared with the control group, treatment of PANC-1 and SUIT-2 tumorspheres with 3.5 μM olaparib suppressed their tumorsphere formation efficacy/viability by 74.2% (*p* < 0.05) and 69.5% (*p* < 0.05), respectively. Treatment with 4.5 μM PD173074 resulted in a 60% (*p* < 0.05) and a 42.9% (*p* < 0.05) reduction in tumorsphere formation efficacy, respectively, whereas exposure to combined PD173074/olaparib treatment reduced the number of formed PANC-1 or SUIT-2 tumorspheres by approximately 91% (*p* < 0.05) (Figure 5A). Consistent with these findings, we also showed that treatment with olaparib and/or PD173074 significantly inhibited the ability of PANC-1 or SUIT-2 cells to form colonies, and this was most apparent in cells exposed to combined therapy, as demonstrated by a 78.3% (*p* < 0.01) or 89.3% (*p* < 0.01) reduction in PANC-1 or SUIT-2 clonogenicity, respectively (Figure 5B). Furthermore, our results indicated that the observed reduction in tumorsphere formation efficiency by olaparib and/or PD173074 was associated with increased RAD51 and γ-H2AX nuclear staining in the PANC-1 or SUIT-2 cells (Figure 5C).

### 3.6. In Vivo, PDAC Cancer Stem Cells are More Sensitive to PD173074 in the Absence of PARP1

To examine the translatability of our in vitro findings, we assessed the efficacy of PD173074 and olaparib and investigated if initial tumor size affects the efficacy of drugs in vivo, and we used tumor xenograft mice models obtained by implantation of dasatinib-resistant PANC-1 tumorsphere cells. The mice were divided into two major treatment groups based on time of treatment initiation, namely “early start” group (*n* = 20, with treatment initiated on Day 3 of Week 1) and “late start” group (*n* = 20, with treatment started on Day 2 of Week 3 with tumor volume ≥ 200 mm^3^) (Figure 6A). In both early and late start treatment groups, the tumor-bearing mice that were treated with 50 mg/kg olaparib combined with 20 mg/kg PD173074 by intraperitoneal injection daily for five consecutive days per week until the end of Week 9 (*n* = 5 per group) exhibited significantly greater tumor growth inhibition than the PBS-treated control group (*n* = 5 per group) or those treated with either 50 mg/kg olaparib (*n* = 5 per group) or 20 mg/kg PD173074 (n = 5 per group) alone (Figure 6B,C). We also observed that the mice with early treatment initiation, regardless of regimen, showed greater inhibition of tumor growth than those with late treatment initiation. Consistent with the trend in tumor growth, the weights of the tumors obtained from mice in the early start group, regardless of regimen, were significantly lesser than those from mice in the late start group (average tumor weight: 0.937 ± 0.526 vs. 1.283 ± 0.817, *p* < 0.05). The mice treated with a combination of PD173074 and olaparib exhibited lesser tumor weights than those from the control subgroup or those treated with either PD173074 or olaparib alone (Figure 6D,E). In addition, we observed that compared with the late start group with 100% tumor xenograft growth, early treatment initiation caused a 35% reduction in tumor xenograft growth, as only 13 out of 20 mice grew tumors, regardless of the treatment regimen (Figure 6F). Furthermore, using immunohistochemical staining, we demonstrated that compared with the PBS-treated control group, tissues from tumors extracted from the BALB/c mice treated with PD173074 and/or olaparib exhibited significantly lower expression of FGFR1, ki-67, and Bcl-2 proteins, with concomitant upregulated cleaved PARP expression (Figure 6G).

## 4. Discussion

Despite improved understanding of PDAC biology alongside numerous diagnostic and therapeutic advancements made in the management of patients with PDAC in the last two decades, the therapeutic success and prognosis of PDAC remain dismal. This necessitates the discovery of new molecular targets and/or the development of novel therapeutic strategies with evidence-based efficacy against PDAC aggression and molecular events implicated in tumor initiation, distant dissemination, innate or acquired resistance to therapy, and therapy failure in patients with PDAC [3,4,19,20,21].

Corroborating the findings of our previous study on the role of aberrant FGFR1 signaling in driving PDAC oncogenicity and stem cell-like phenotype, the therapeutic targetability of FGFR1, and the anti-PDAC efficacy of PD173074, a selective FGFR1 inhibitor [13], the present study provided better mechanistic insight into the therapeutic exploitability of aberrantly expressed FGFR1 in PDAC. First, we demonstrated that enhanced FGFR1 and PARP1 expression characterized FGFR1 inhibitor-resistant pancreatic cells at mRNA and protein levels. Secondly, we showed that the observed aberrant FGFR1 and PARP1 expression was associated with cancer stem cell-like phenotype, regulated DNA repair, and modulated the response to therapy in PDAC cells. Thirdly, our data indicated that FGFR1 and PARP1 co-occurred and directly interacted with each other at the expense of RAD51/H2AFX signaling and had adverse prognostic implications. Fourthly, we provided preclinical evidence that olaparib-induced pharmacological inhibition of PARP1 synergistically enhanced the therapeutic effect of the FGFR1 inhibitor PD173074. Fifthly, we demonstrated that olaparib alone or synergistically with PD173074 suppressed the oncogenic cancer stem cell-like phenotype of PDAC cells through the apoptosis-related impairment of DNA repair. Finally, our results showed that PDAC cancer stem cells were more sensitive to the FGFR1 inhibitor PD173074 in the absence of PARP1 in vivo. These findings improve our current understanding of PDAC biology and may help inform future therapeutic decision-making in the management of patients with PDAC, considering the present clinical challenge of therapy failure and poor prognosis in PDAC cohorts.

To the best of our knowledge, the present study is the first to demonstrate that resistance to the FGFR1 inhibitor, dasatinib, is partly associated with enhanced FGFR1 and PARP1 expression at mRNA and protein levels (Figure 1). This finding was corroborated by previous findings indicating that FGFR1 amplification was significantly associated with overexpression of FGFR1, drove anchorage-independent cell proliferation, and enhanced resistance to endocrine therapy [22,23], as well as mediated resistance to cyclin-dependent kinase 4/6 inhibitors in breast cancer [24]. Moreover, our findings were corroborated partly by the results of another study, indicating that the inhibition of PARP-1 activity was elicited by induction of the tumor necrosis factor-related apoptosis-inducing ligand (TRAIL) signaling with activation of caspase-3 and -8, which sensitized TRA-8-resistant PANC-1 and SUIT-2 cells to the apoptotic action of a monoclonal agonist antibody, which specifically targeted death receptor 5 [25]. Although our present study did not address the role of caspase-8 in the re-sensitization of resistant PANC-1 and SUIT-2 cells to therapy, we believe that though the induction of DNA damage was not necessary for the induction of cell death or apoptosis by death ligands (DLs), DLs elicited DNA damage in surviving cancerous cells. Thus, based on the findings of Yuan et al., we suggest that FGFR1- and PARP-rich pancreatic cancer cells maintain their insensitivity to single-agent FGFRs and evade cell death by suppressing TRAIL-associated apoptotic DNA fragmentation by caspase-activated DNase/DNA fragmentation factor 40 endonuclease, thus suggesting a critical role of DNA damage/repair balance in pancreatic cancer therapy response [26,27].

We demonstrated that aberrant FGFR1 and PARP1 expression was associated with cancer stem cell-like phenotype, regulated DNA repair, and modulated response to therapy in PDAC cells. We also reported that the upregulated nested cluster of FGFR1/PARP1/ALDH1A1/ABCB1 was inversely correlated to and diametrically distant from the downregulated cluster of H2AFX/RAD51 (Figure 2). These findings were consistent with our current understanding of the critical roles of the cancer stem cell marker ALDH1A1 and the multidrug resistance protein ATP binding cassette subfamily B member 1 (ABCB1/MDR1) in the enhancement of tumorigenicity, activation of DNA repair, and resistance to anticancer therapy [28]. Moreover, our findings are particularly relevant when viewed in the context of recently published work indicating that aberrant FGFR1 expression in the tumor vascular niche was implicated in the transformation of indolent cancer cells into chemoresistant cancer stem cells and that FGFR1+ cells were associated with the engraftment of chemoresistant cancer stem cells [29]. Similarly, PARP1 has been shown to modulate sensitivity to therapy, and olaparib-based therapy significantly enhanced the sensitivity of pancreatic cancer cells (particularly the p53 mutant cells) and other isogenic cancer cell lines to radiotherapy, by inducing cell cycle arrest in G2, homologous recombination repair inhibition, and persistent DNA damage responses [30].

The implication of FGFR1, PARP1, ALDH1A1, ABCB1, RAD51, and H2AFX in therapy response was highlighted by our results. The patients with PDAC were stratified into therapy responders and non-responders. *Gene set_high* patients exhibiting high FGFR1, PARP1, ALDH1A1, and ABCB1 expression, but low RAD51 and H2AFX expression levels were less sensitive to erlotinib and gemcitabine compared to the *Gene set_low* patients who had low FGFR1, PARP1, ALDH1A1, and ABCB1 expression, but high RAD51 and H2AFX expression level (Figure 2). We posit that the relative non-responsiveness of the *Gene set_high* patients may be related to their low mutagenicity, a constitutively active DNA damage response, but impaired replication stress response due to probable suppressed levels of phospho-H2AX and RAD51 at baseline [31]; this hypothesis was consistent with our results showing that FGFR1 and PARP1 co-occurred and directly interacted with each other at the expense of RAD51/H2AFX signaling and with adverse prognostic implications (Figure 3). However, a limitation to this current study remains. It has been established that FOLFIRINOX followed by PARP inhibitor (olaparib) was a maintenance therapy for BRCA mutant pancreatic cancer patients. We have yet to determine if FGFR inhibition could result in the downregulation of BRA1/2 and PALB2, since the synthetic lethal interaction between HR deficiency and PARP inhibition has been shown. This issue is currently being investigated in our laboratory to provide further insights into the mechanistic evidence.

Consistent with the understanding that exposure to PARP inhibitors can enhance the burden of unrepaired DNA double-strand breaks (DSBs) by impeding PARP1 activity and PARP1 trapping onto damaged DNA [32], we demonstrated that olaparib-induced pharmacological inhibition of PARP1 synergistically enhanced the therapeutic effect of the FGFR1 inhibitor PD173074 and that olaparib alone or synergistically with PD173074 suppressed the oncogenic cancer stem cell-like phenotype of PDAC cells through the apoptosis-related impairment of DNA repair (Figure 4 and Figure 5). These results are consistent with recent findings that impairments in DNA repair and accumulation of lethal DNA DSBs induced by tyrosine kinase inhibitors sensitize quiescent and proliferative acute myeloid leukemia stem cells to PARP inhibitors olaparib and BMN673 [33], indicating synthetic lethality, a biological phenomenon wherein cell death is induced more efficiently by the simultaneous loss of function of multiple molecular targets or genes compared with the loss of function of a single gene [34]. Similarly, treatment with pan-kinase inhibitors or selective RTKi, namely HS10241, with the PARPi fluzoparib synergistically suppressed tumor growth in multiple cancer types both in vitro and in vivo [32], which is consistent with our findings indicating that PDAC cancer stem cells were more sensitive to the FGFR1 inhibitor PD173074 in the absence of PARP1, in vivo (Figure 6).

## 5. Conclusion

The present study (Figure 7) was the first, to the best of our knowledge, to demonstrate that olaparib + PD173074 combined therapy was synthetically lethal to treatment-resistant PDAC cells. The selective inhibition of PARP1 by olaparib synergized with exposure to PD173074 to elicit significant suppression of PDAC stem cell viability and related cancer stem cell-like phenotypes. The synthetic lethality induced by combining olaparib with PD173074 and the anti-PDAC efficacy were in part mediated by enhanced RAD51 and γ-H2AX expression and/or activity. This preclinical study provided the foundation for further exploration of the clinical applications of olaparib + PD173074 combination therapy-induced synthetic lethality in PDAC.

## Figures and Tables

**Figure 1 cells-09-00911-f001:**
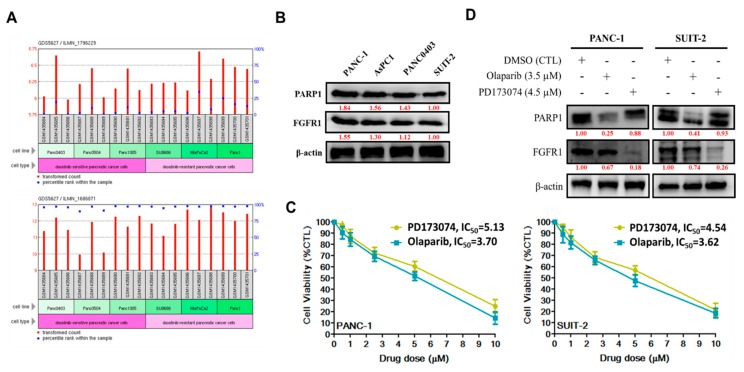
Enhanced FGFR1 and PARP1 expression characterizes FGFR1 inhibitor-resistant pancreatic cells at mRNA and protein levels. (**A**) Histograms of the FGFR1 (upper panel) and PARP1 (lower panel) RNAseq expression profile in dasatinib-sensitive PANC0403, PANC0504, and PANC1005 cells compared with dasatinib-resistant SU8686, MiaPaCa2, and PANC-1 pancreatic cancer cells in the GSE9357/GP10558/GDS5627 cohort. (**B**) Representative Western blot data showing the differential expression of FGFR1 and PARP in PANC-1, AsPC1, PANC0403, or SUIT-2 cells. (**C**) Line graphs showing the effect of 1–10 μM PD173074 or olaparib on the viability of PANC-1 (left panel) or SUIT-2 (right panel) cells. (**D**) Representative Western blot images of the effect of 4.5 μM PD173074 or 3.5 μM olaparib on the expression level of PARP or FGFR1 protein in the PANC-1 and SUIT-2 cells. β-actin served as the loading control.

**Figure 2 cells-09-00911-f002:**
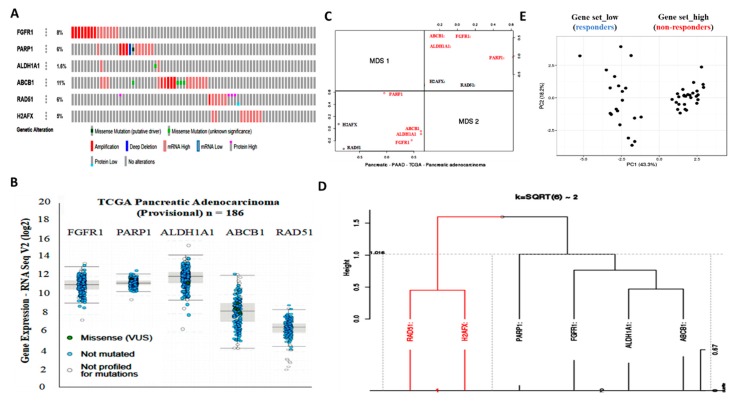
Aberrant FGFR1 and PARP1 expression is associated with cancer stem cell-like phenotype, regulates DNA repair, and modulates response to therapy in PDAC cells. (**A**) Oncoprint analysis of FGFR1, PARP1, ALDH1A1, ABCB1, RAD51, and H2AFX genetic alterations in the provisional TCGA pancreatic adenocarcinoma cohort. (**B**) Expression profile and (**C**) multidimensional scaling correlation matrix with (**D**) k-means clustering of FGFR1, PARP1, ALDH1A1, ABCB1, RAD51, and H2AFX mRNA in the provisional TCGA pancreatic adenocarcinoma cohort. (**E**) Gene-set expression-based cohort stratification into therapy responders and non-responders using the E-GEOD-17891, AFFY_HG_U133_PLUS_2 dataset (*n* = 47). Clustering was based on correlation distance and average linkage. Unit variance scaling is applied to rows; Singular-value decomposition with imputation was used for principal component (PC) calculation. X and Y axes show PC 1 and PC 2, which explain 43.3% and 18.2% of the total variance, respectively. Missense (VUS): Missense variants of uncertain significance

**Figure 3 cells-09-00911-f003:**
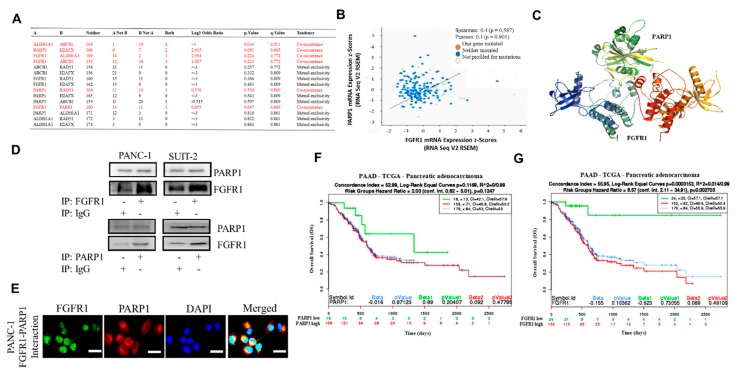
FGFR1 and PARP1 co-occur and directly interact with each other at the expense of RAD51/H2AFX signaling and with adverse prognostic implications. (**A**) Co-occurrence and mutual exclusivity analysis of FGFR1, PARP1, ALDH1A1, ABCB1, RAD51, and H2AFX in the TCGA pancreatic adenocarcinoma cohort. (**B**) FGFR1-PARP1 gene expression correlation in the TCGA pancreatic adenocarcinoma cohort. (**C**) Schrödinger’s PyMOL molecular graphics system-generated molecular docking of PARP1 and FGFR1. (**D**) PANC-1 and SUIT-2 cells were subjected to immunoprecipitation (IP) with FGFR1 (upper) or PARP1 (lower) antibody followed by Western blotting analysis with the indicated antibodies, with IgG serving as the control. (**E**) Interaction of PARP1 with fibroblast growth factor receptor 1 (FGFR1). Coimmunolocalization assays of PARP1 and FGFR1 in the PANC1 cancer cell lines expressing wild-type PARP1 and FGFR1. Kaplan–Meier plots showing the differential overall survival (OS) between (**F**) FGFR1-low and FGFR1-high samples or (**G**) PARP1-low and PARP1-high samples from the TCGA pancreatic adenocarcinoma cohort.

**Figure 4 cells-09-00911-f004:**
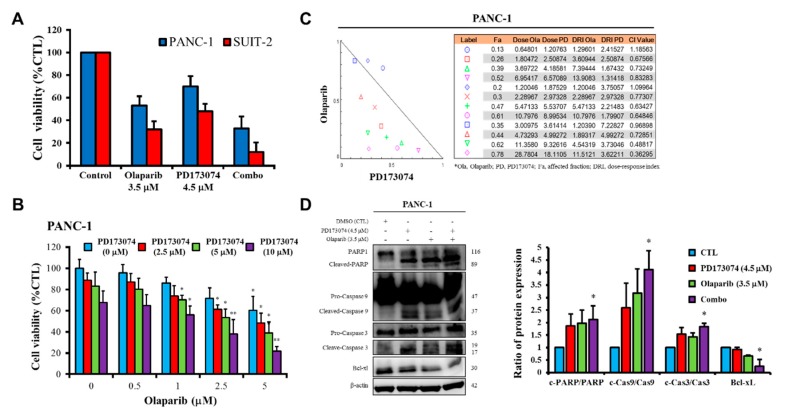
Olaparib-induced pharmacological inhibition of PARP1 synergistically enhances the therapeutic effect of PD173074. (**A**) Graphical representation of changes in PANC-1 and SUIT-2 cell viability when treated with 4.5 μM PD173074 and/or 3.5 μM olaparib for 48 h. (**B**) Effect of combining 2.5–10 μM PD173074 and 0.5–5 μM olaparib on the viability of PANC-1 cells. (**C**) Isobologram and drug combination index plot indicating synergy between different concentrations of PD173074 and olaparib. (**D**) The effect of treatment with 4.5 μM PD173074 and/or 3.5 μM olaparib on the expression level of PARP, cleaved PARP, pro-caspase-9, cleaved-caspase-9, pro-caspase-3, cleaved-caspase-3, and Bcl-xL proteins as shown by Western blot analysis. * *p* < 0.05, ** *p* < 0.01, (PANC-1: treated vs. CTL); * *p* < 0.05, ** *p* < 0.01 (SUIT-2: treated vs. CTL); CTL, control; Combo, PD173074 + olaparib combination therapy; Fa, fraction affected; CI, combination index; DRI, dose-reduction index.

**Figure 5 cells-09-00911-f005:**
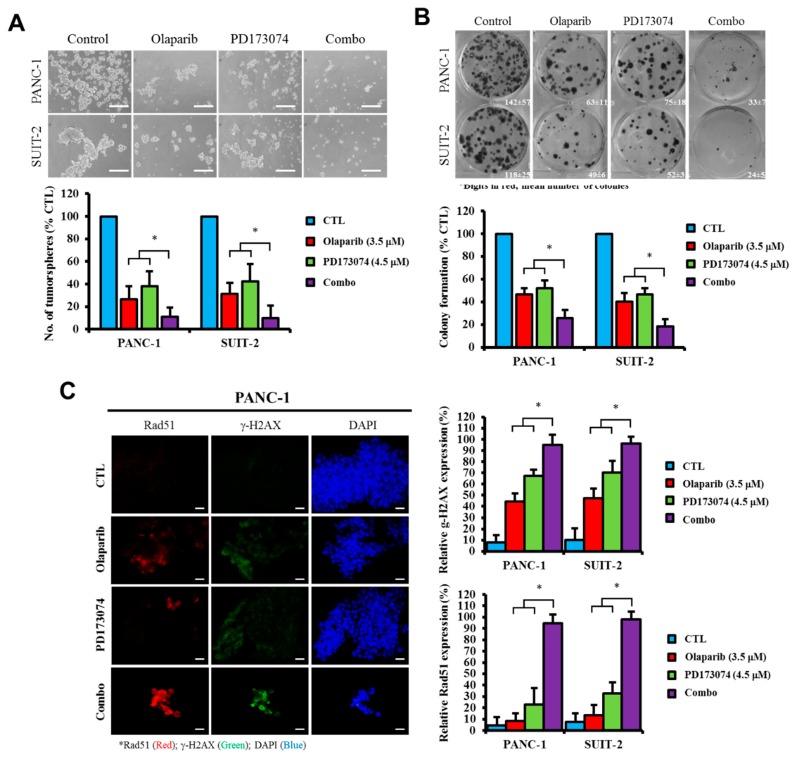
Olaparib alone or in synergism with PD173074 suppresses the oncogenic cancer stem cell-like phenotype of PDAC cells through an apoptosis-related impairment of DNA repair. (**A**) Tumorsphere formation assay images and histograms showing the effect of PD173074 and/or olaparib on the tumorsphere formation efficiency of the PANC-1 or SUIT-2 cells. (**B**) Representative colony formation assay data (left) and graph (right) showing the effect of PD17374 and/or olaparib on the number of colonies formed by PANC-1 or SUIT-2 cells. (**C**) Immunofluorescence images and histograms of γ-H2AX (red), RAD51 (green), and nuclear DAPI (blue) staining in PANC-1 cells treated with PD173074 and/or olaparib compared with the control. Scale bar, 10 μm. (**D**) Downregulation of FGFR1, Sox2, and Nanog protein expression levels in PD173074-treated Panc-1 cells as determined by Western blot analysis. β-actin was used as a loading control. * *p* < 0.05

**Figure 6 cells-09-00911-f006:**
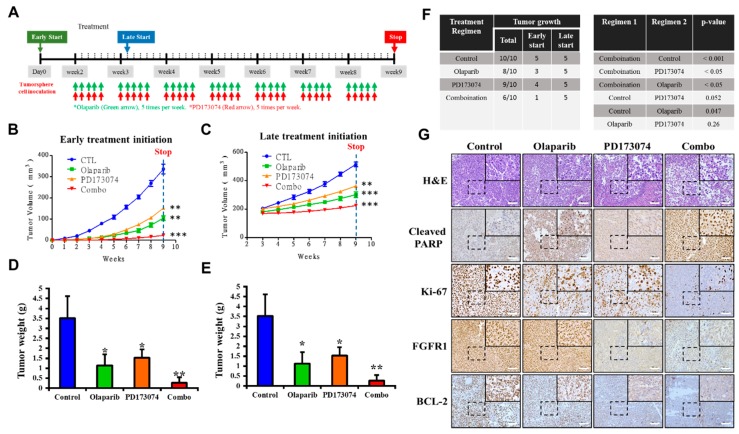
In vivo, PDAC cancer stem cells are more sensitive to PD173074 in the absence of PARP1. (**A**) Experimental chart of in vivo studies. Line graphs of the differential effect of (**B**) early or (**C**) late initiation of treatment with PD173074 and/or olaparib on BALB/c mice bearing PANC-1 tumorsphere-derived tumors. Graphical representation of tumor weight at the end of the experiment for (**D**) early and (**E**) late treatment initiation groups. (**F**) Frequency of tumor growth and correlative statistical analysis of treatment regimen in the PD173074- and/or olaparib-treated mice. (**G**) Representative immunohistochemical staining images showing differential expression of cleaved PARP, Ki-67, FGFR1, and Bcl-2 in tumors extracted from the PD173074- and/or olaparib-treated NOD/SCID mice. * *p* < 0.05; ** *p* < 0.01; *** *p* < 0.001.

**Figure 7 cells-09-00911-f007:**
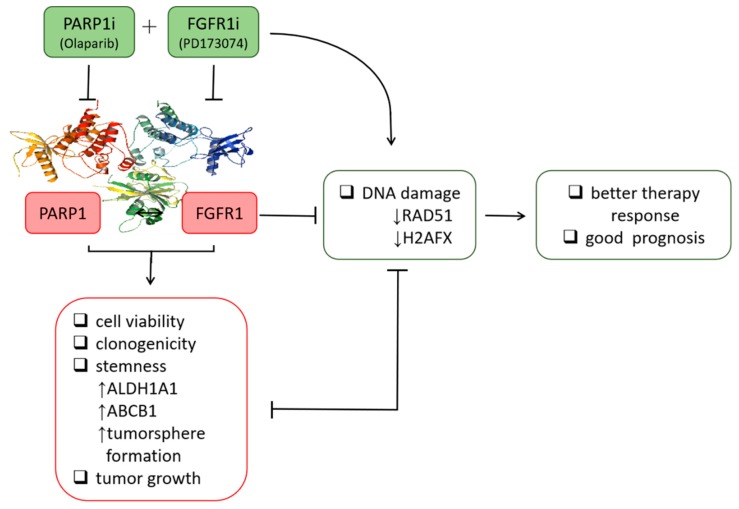
Schematic abstract showing targeted inhibition of PARP1 combined with FGFR1 blockade concomitantly suppresses DNA repair and inhibits the viability, colony formation, and tumorsphere formation of resistant pancreatic cells. Furthermore, suppressed tumor growth in vivo indicates the synthetic lethality of the dual PARP1/FGFR1 inhibition to malignant cells in patients with pancreatic cancer.

## Data Availability

The datasets used and analyzed in the current study will be made available by the corresponding author in response to reasonable requests.

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
