# Peer review of "Targeted PARP Inhibition Combined with FGFR1 Blockade is Synthetically Lethal to Malignant Cells in Patients with Pancreatic Cancer"

_cells, 2020, doi:10.3390/cells9040911_

Round 1
Reviewer 1 Report
The manuscript continues to be difficult to read both in the text and figures. It is too long. Conclusions of the study are not clearly supported by the results and its application in the clinical practice is questionable (results of studies with olaparib in advanced pancreatic cancer are disappointing).
In the section Introduction Authors stated that the standard of care for pancreatic cancer is gemcitabine with paclitaxel, erlotinib or capecitabine, and in the next paragraph the current standard of care for metastatic disease is FOLFIRINOX. These sentences are apparently in contrast.
Author Response
Response to Reviewers:
Point-by-point responses to reviewer’s comments - Reviewer 1:
We thank the reviewer for carefully reading our manuscript and providing valuable comments. We believe making use of all these comments has further helped improve the quality and appeal of our work, as well as strengthened the manuscript. Below are our point-by-point responses.
Q1.1. The manuscript continues to be difficult to read both in the text and figures. It is too long. Conclusions of the study are not clearly supported by the results and its application in the clinical practice is questionable (results of studies with olaparib in advanced pancreatic cancer are disappointing).
A1.1. A1: We sincerely thank the reviewer for this important comment. This document certifies that the manuscript listed below was edited for proper English language, grammar, punctuation, spelling, and overall style by one or more of the highly qualified natives English speaking editors at American Journal Experts. Please kindly see our revised manuscript.
Q1.2. In the section Introduction Authors stated that the standard of care for pancreatic cancer is gemcitabine with paclitaxel, erlotinib or capecitabine, and in the next paragraph the current standard of care for metastatic disease is FOLFIRINOX. These sentences are apparently in contrast.
A1.2. We sincerely thank the reviewer for these comments. While apologizing for the alluded lack of clarity by the reviewer, we humbly point out that there is no apparent conflict in the paragraph alluded to, as in oncology clinics, the treatment modality for primary disease may differ from that for advanced or metastatic disease. To address the reviewer’s concern, we have modified the sentence in our revised manuscript. Please kindly see our revised Introduction section, page 2 Lines 68-89 to page 3, line 1-35.
Pancreatic cancer is the seventh leading cause of cancer-related deaths globally; it accounts for more than 20% of all deaths caused by abdominal cancers and approximately 5% of all cancer-related deaths [1]. Most deaths are histologically attributed to the pancreatic ductal adenocarcinoma (PDAC) subtype, which originates from the pancreatic duct epithelium, represents approximately 85% of the total incidence of pancreatic cancer, and is often unresectable; consequently, patients have limited treatment options and exhibit high morbidity and mortality rates [1, 2]. A yearly increase was reported in PDAC-specific mortality rates (approximately 98% mortality/incidence), and a consistently low 5-year survival rate of <5% was reported over the past decade [1].
Currently, the standard of care is gemcitabine with nanoparticle albumin-bound-paclitaxel, erlotinib, or capecitabine, while for advanced stage (III and IV) or metastatic PDAC, the preferred chemotherapy is a combination of fluorouracil, leucovorin, irinotecan, and oxaliplatin known as FOLFIRINOX; However, these chemotherapeutic drugs are associated with a high risk of severe drug-related toxicity, acquired resistance, and non-significant survival benefits [3, 4]. Hence, the discovery of new actionable molecular oncotargets is necessary. Furthermore, screening methods should be developed for identifying novel and highly efficacious therapeutic agents that inhibit disease progression, alleviate resistance to treatment, and improve prognosis in patients with PDAC.
The involvement of highly conserved and ubiquitous transmembrane tyrosine kinase receptors, namely fibroblast growth factor receptors (FGFRs), in tumorigenesis, cell fate determination, survival, motility, angiogenesis, and malignantization of tumor cells, as well as in reduced sensitivity to anticancer therapeutics and poor prognosis, are extensively documented [5]. Notably, genetic alterations in FGFR1, such as amplification or an increase in gene copy number, are positively correlated with overexpression, and are more prevalent than genetic aberrations in FGFR2, FGFR3, and FGFR4 [5, 6]. Hence, Lehnen et al studying the role of the FGFR1 gene copy number and expression pattern in patients with PDAC suggested that the association between FGFR1 amplification, mRNA or protein expression, and the proliferative potential of PDAC cells can be exploited for therapeutic purposes using FGFR1 inhibitors [7]. This is consistent with results indicating that enhanced FGFR signaling predisposed pancreatic cancer cells to the killing effect of dovitinib, a potent FGFR inhibitor, in preclinical models [8], thus indicating that the pharmacological or molecular targeting of FGFR1 in patients with PDAC has therapeutic potential. However, despite the actual or predicted therapeutic benefits of FGFR1-targeting therapeutics, the post-treatment development of resistance due to accrue de novo genetic alterations in the kinase targets of these therapeutics following extended exposure to FGFR1 inhibitors is extensively documented [9, 10, 11]. Therefore, the discovery or development of adjuvant or maintenance therapeutic agents for treating patients with PDAC harboring FGFR1 aberrations is necessary.
Poly (ADP ribose) polymerase (PARP) plays a critical role in the sensing of genomic damage signals, DNA repair and replication, inflammation, transcriptional and posttranscriptional gene expression modulation, and cell death regulation. PARP has also been implicated in several pathological processes, including carcinogenesis, by PARylation, direct or mediated interactions with oncogenes, and regulation of oncogenic transcription factors [12]; Thus, the increased interest in and exploration of the therapeutic potential and clinical feasibility of PARP inhibition as an effective anticancer strategy in the last decade. Several PARP inhibitors are being developed or currently undergoing clinical evaluation. As the concept of synthetic lethality in anticancer therapy continues to gain traction, the probable exploitability of the DNA repair –disrupting potential of PARP inhibitors (PARPi) is receiving a second look. In the context of the present study, synthetic lethality is implied if the impairment of either of two oncogenic pathways is not lethal or sufficiently detrimental to the viability of cancerous cells, but their combination induces lethality in the cancerous cells. An increasing amount of preclinical and clinical evidence suggests that PARPi may potentiate the effect of conventional chemotherapeutics, enhance sensitivity to radiation therapy, and serve as adjuvants or maintenance therapeutic agents [12].
Against the background of increasing incidence of acquired resistance to the FGFR1 inhibitor and our previous work that partially demonstrates the caspase-3/PARP-mediated anticancer and anti-metastasis efficacy of PD173074, a selective FGFR1 inhibitor, against aldehyde dehydrogenase (ALDH)-rich or FGFR1-high PDAC cells [13], the present study investigates the probable synthetic lethality and therapeutic efficacy of targeted combinatorial FGFR1/PARP inhibition in patients with PDAC harboring FGFR1 aberrations using a multifaceted approach, including bioinformatics-aided prediction and visualization, drug screening, molecular biology, and functional assays.

Reviewer 2 Report
Lai et al. showed the synergistic effect of PARP inhibitor (Olaparib) and FGFR1 inhibitor in two pancreatic cancer cell lines. Additionally, the authors showed a synergistic effect in Panc1 xenograft mouse model. Synthetic lethal interaction between PARPi and FGFRi is interesting. However, a mechanistic insight as to how these two drugs yield a synergistic effect should be added. The authors should have asked whether FGFRi induces a BRCAness by suppressing other homologous recombination pathway genes, as it has been reported that HR-defective pancreatic cancer cells respond to PARP inhibition. Association based data from publicly available data set in Figure 1, 2 and 3 do not seem to add much to the general outline of this story. Co-IP experiments in Figure 3D are even erroneous. How can IgG and negative control IP lanes can have immunoprecipitated protein bands? Moreover, PARP1 is a nuclear protein and FGFR is a membrane protein. What was a rationale to perform Co-IP experiment here? How this interaction can be important for the synergistic effect of PARPi and FGFRi?
In addition, there are typos and errors throughout manuscript. It is recommended to proof-read thoroughly before submission.
To sum, the authors observed an interesting synergism between PARPi and FGFRi. But it lacks a rationale how these two drugs were combined, and the underlying mechanism of drug synergism should have been tested.
Author Response
Point-by-point responses to reviewer’s comments - Reviewer 2:
We would like to thank the reviewer for the thorough reading of our manuscript as well as their valuable comments. We believe all comments are borne out of good faith, and thus, have tried to address their comments conscientiously and feel that they have further improved the readability and appeal of our work, as well as strengthened the manuscript. Below are our point-by-point responses.
Q2.1. Lai et al. showed the synergistic effect of PARP inhibitor (Olaparib) and FGFR1 inhibitor in two pancreatic cancer cell lines. Additionally, the authors showed a synergistic effect in Panc1 xenograft mouse model. Synthetic lethal interaction between PARPi and FGFRi is interesting.
A2.1. We thank the reviewer for taking time to read our manuscript and for the critiques and suggestions made in order to help us improve the quality of our work. In this revised manuscript, we have made effort to make address all the comments and suggestions.
Q2.2. However, a mechanistic insight as to how these two drugs yield a synergistic effect should be added.
A2.2. We thank the reviewer for this comment. As suggested by the reviewer, we have now improved the clarity of the section providing “a mechanistic insight as to how these two drugs yield a synergistic effect” in our revised manuscript. Please kindly see our revised Results section, pages 9, lines 20-34 to page 10, line 1-13.
3.4. Olaparib-induced pharmacological inhibition of PARP1 expression synergistically enhances the therapeutic effect of FGFR1 inhibitor PD173074 through a caspase-dependent apoptotic mechanism
Having shown that FGFR1 and PARP1 co-occur and directly interact with each other at the expense of RAD51/H2AFX signaling, as well as adversely affect prognosis, we examined whether the FGFR1/PARP1 interaction could be exploited to enhance the anticancer effect of FGFR1 inhibition or overcome resistance to the latter. Our results showed that single-agent treatment with 3.5 μM olaparib or 4.5 μM PD173074 resulted in 47% or 30% loss of PANC-1 cell viability, respectively (p < 0.05), and 68% or 52% loss of SUIT-2 cell viability, respectively (p < 0.05). However, combination treatment with olaparib/PD173074 induced a 67% (p < 0.01) reduction in the viability of PANC-1 cells and 89% (p < 0.01) reduction in SUIT-2 cell viability (Figure 4A). Furthermore, we showed that 0.5–5 μM olaparib dose-dependently enhanced the killing effect of PD173074 in PANC-1 cells, with 1 μM olaparib increasing the PANC-1 cell-killing effect of 2.5, 5, and 10 μM PD173074 by 16%, 13%, and 12%, respectively; while a 39%, 42%, and 44% increase in the cytotoxic effects of 2.5, 5, and 10 μM PD173074, respectively, was observed when combined with 5 μM olaparib (Figure 4B). To determine whether olaparib enhanced the effect of PD173034 through a synergistic or additive mechanism, we performed drug combination analyses by using Chou-Talalay’s algorithm-based isobologram to evaluate dual-agent inhibitory effects and combination indices. As shown by the isobologram, all combination dose-points, except two, were within the right-angled isobologram triangle, indicating olaparib–PD173074 synergism in the PANC-1 cells (Figure 4C). In parallel experiments to elucidate the mechanism underlying the observed anticancer effect of olaparib and/or PD173074, we demonstrated that compared with the mild-to-moderate upregulation of cleaved PARP/PARP, cleaved-caspase (CASP)9/CASP9, and cleaved-CASP3/CASP3 ratios in PANC-1 cells treated with 4.5 μM PD173074 or 3.5 μM olaparib alone, combining PD173074 with olaparib significantly increased the cleaved PARP/PARP, cleaved- CASP9/CASP9, and cleaved-CASP3/CASP3 ratios, concomitantly with a marked decrease in the expression level of Bcl-xL protein (Figure 4D). Thus, indicating, at least in part, that the synergistic effect of PD173074 and olaparib is caspase-dependent and mediated by activation of the CASP3/CASP9/PARP cell death signaling.
Please also kindly see our newly Figure 4 and its legend, page 10, lines 14-25.
Figure 4. Olaparib-induced pharmacological inhibition of PARP1 synergistically enhances the therapeutic effect of PD173074. (A) Graphical representation of changes in PANC-1 and SUIT-2 cell viability when treated with 4.5 μM PD173074 and/or 3.5 μM olaparib for 48 hours. (B) Effect of combining 2.5–10 μM PD173074 and 0.5–5 μM olaparib on the viability of PANC-1 cells. (C) Isobologram and Drug Combination Index Plot indicating synergy between different concentrations of PD173074 and olaparib (D) The effect of treatment with 4.5 μM PD173074 and/or 3.5 μM olaparib on the expression level of PARP, cleaved PARP, pro-caspase 9, cleaved-caspase 9, pro-caspase 3, cleaved-caspase 3, and Bcl-xL proteins as shown by Western blot analysis. *p < 0.05, **p < 0.01, ***p < 0.001 (PANC-1: treated vs CTL); §p < 0.05, §§p < 0.01, §§§p < 0.001 (SUIT-2: treated vs CTL); CTL, control; Comb, PD173074 + Olaparib combination therapy; Fa, fraction affected; CI, combination index; DRI, dose-reduction index.
Q2.3. The authors should have asked whether FGFRi induces a BRCAness by suppressing other homologous recombination pathway genes, as it has been reported that HR-defective pancreatic cancer cells respond to PARP inhibition.
A2.3. We sincerely thank the reviewer for this suggestion. We do agree with the reviewer that it was recently shown by Keung MYT, et al. that PARP inhibitors may serve as therapeutic agents for homologous recombination deficiency in breast cancers [J Clin Med. 2019; 8(4):435], and that TGFb induces “BRCAness” and sensitivity to PARP inhibition in breast cancer by regulating DNA-repair genes [Liu L, et al. Mol Cancer Res. 2014; 12(11):1597-609], however, the present study based on accruing evidence on the anticancer efficacy of synthetic lethality (SL) in various cancer types, explored the therapeutic feasibility of targeting PARP in combination with FGFR1 as an efficacious SL-based anti-cancer strategy for patients with pancreatic cancer.
Q2.4. Association based data from publicly available data set in Figure 1, 2 and 3 do not seem to add much to the general outline of this story. Co-IP experiments in Figure 3D are even erroneous. How can IgG and negative control IP lanes can have immunoprecipitated protein bands? Moreover, PARP1 is a nuclear protein and FGFR is a membrane protein. What was a rationale to perform Co-IP experiment here? How this interaction can be important for the synergistic effect of PARPi and FGFRi?
A2.4. We are very grateful for the reviewer’s comments.
In response to the reviewer’s question that “PARP1 is a nuclear protein and FGFR is a membrane protein. What was a rationale to perform Co-IP experiment here?”, may we humbly remind the erudite reviewer that the subcellular localization of FGFR1 (just like EGFR) is not absolute; in fact it is evidence-based that FGFR1 translocalizes to and is enriched in the nuclei of cells with same confidence as in the membrane. Please kindly see https://www.genecards.org/cgi-bin/carddisp.pl?gene=FGFR1:
Also kindly see https://www.uniprot.org/uniprot/P11362#subcellular_location
In addition, with regards to the “…rationale to perform Co-IP experiment…”, as with most studies of this nature, co-IP in this study was used to identify or validate protein (FGFR1)-protein (PARP1) interaction. Please kindly see: John T. Corthell Ph.D., in Basic Molecular Protocols in Neuroscience: Tips, Tricks, and Pitfalls, 2014 Chapter 8 -Immunoprecipitation: pp 77-81. http://sci-hub.tw/10.1016/B978-0-12-801461-5.00008-3
Further to address the reviewer’s concern, we have now provided a more representative data for newly Figure 3D in our revised manuscript. Please kindly see our revised Results section, pages 8, lines 18-32 to page 9, line 1-7.
3.3. FGFR1 and PARP1 co-occur and directly interact with each other at the expense of RAD51/H2AFX signaling and exhibit adverse prognostic implications
To further characterize the functional association between FGFR1, PARP1, ALDH1A1, ABCB1, RAD51, and H2AFX, as well as to provide a therapeutic context, we performed a co-occurrence and mutual exclusivity analysis using the cBioPortal for cancer genomics platform (https://www.cbioportal.org/). Our results indicate that while FGFR1, PARP1, ALDH1A1, and ABCB1 exhibited varying degrees of co-occurrence, their occurrence was mutually exclusive to that of RAD51 or H2AFX (Figure 3A). Reanalysis of the TCGA-PAAD cohort (n = 176) showed moderate positive correlation between FGFR1 and PARP1 mRNA expression (Spearman’s R: 0.4; Pearson’s R: 0.1) (Figure 3B). Furthermore, using Schrödinger’s PyMOL molecular graphics system (https://pymol.org/2/) for spatiotemporal visualization of probable molecular interaction between FGFR1 and PARP1, we demonstrated that FGFR1 (protein data bank, PDB: 3C4F) directly binds with PARP1 (PDB: 5DS3) with a complementarity score of 18008 to form a stable complex with an interface area of 2482.7Å2 and atomic contact energy (ACE) of 286.24 kcal/mol (Figure 3C). In parallel assays, consistent with the in silico data, results of our co-immunoprecipitation analyses demonstrated direct interaction between FGFR1 and PARP1 in FGFR1-reconstituted PANC-1 and SUIT-2 cells compared with their IgG-treated control counterparts (Figure 3D). In addition, we demonstrated that compared with patients with low PARP1 expression, those with high PARP1 expression exhibited a 1.69-fold worse overall survival (OS) with a mortality index of 0.962 and hazard ratio (HR) = 2.03 (confidence interval, CI: 0.82–5.01) (Figure 3E). Compared to patients with low FGFR1 expression, those with high FGFR1 expression exhibited about 5.25-fold worse OS with a mortality index of 0.974 HR of 8.57 (CI: 2.11–34.91) (Figure 3F).
Please also see our newly Figure 3D and its legend, Page 9, Lines 9-18.
Figure 3. FGFR1 and PARP1 co-occur and directly interact with each other at the expense of RAD51/H2AFX signaling and with adverse prognostic implications. (A) Co-occurrence and mutual exclusivity analysis of FGFR1, PARP1, ALDH1A1, ABCB1, RAD51, and H2AFX in the TCGA Pancreatic Adenocarcinoma cohort. (B) FGFR1 - PARP1 gene expression correlation in the TCGA Pancreatic Adenocarcinoma cohort. (C) Schrödinger’s PyMOL molecular graphics system-generated molecular docking of PARP1 and FGFR1. (D) PANC-1 and SUIT-2 cells were subjected to immunoprecipitation (IP) with FGFR1 (upper) or PARP1 (lower) antibody followed by Western blotting analysis with the indicated antibodies, and IgG serving as control. Kaplan–Meier plots showing the differential overall survival (OS) between (E) FGFR1-low and FGFR1-high samples, or (F) PARP1-low and PARP1-high samples from the TCGA Pancreatic Adenocarcinoma cohort.
Q2.5. In addition, there are typos and errors throughout manuscript. It is recommended to proof-read thoroughly before submission.
A2.5. We thank the reviewer for this very important comment. We have now taken time to carefully revise our manuscript for likely errors in English language grammar or typography. We believe this has helped improve the clarity and comprehensibility of our revised manuscript. Please kindly see our revised manuscript.
Q2.6. To sum, the authors observed an interesting synergism between PARPi and FGFRi. But it lacks a rationale how these two drugs were combined, and the underlying mechanism of drug synergism should have been tested.
A2.6. We are grateful for the reviewer’s comment. We do believe the reviewer’s concern has now been addressed in our revised manuscript. Please kindly see our revised Results section, pages 9, lines 20-34 to page 10, line 1-13.
3.4. Olaparib-induced pharmacological inhibition of PARP1 expression synergistically enhances the therapeutic effect of FGFR1 inhibitor PD173074 through a caspase-dependent apoptotic mechanism
Having shown that FGFR1 and PARP1 co-occur and directly interact with each other at the expense of RAD51/H2AFX signaling, as well as adversely affect prognosis, we examined whether the FGFR1/PARP1 interaction could be exploited to enhance the anticancer effect of FGFR1 inhibition or overcome resistance to the latter. Our results showed that single-agent treatment with 3.5 μM olaparib or 4.5 μM PD173074 resulted in 47% or 30% loss of PANC-1 cell viability, respectively (p < 0.05), and 68% or 52% loss of SUIT-2 cell viability, respectively (p < 0.05). However, combination treatment with olaparib/PD173074 induced a 67% (p < 0.01) reduction in the viability of PANC-1 cells and 89% (p < 0.01) reduction in SUIT-2 cell viability (Figure 4A). Furthermore, we showed that 0.5–5 μM olaparib dose-dependently enhanced the killing effect of PD173074 in PANC-1 cells, with 1 μM olaparib increasing the PANC-1 cell-killing effect of 2.5, 5, and 10 μM PD173074 by 16%, 13%, and 12%, respectively; while a 39%, 42%, and 44% increase in the cytotoxic effects of 2.5, 5, and 10 μM PD173074, respectively, was observed when combined with 5 μM olaparib (Figure 4B). To determine whether olaparib enhanced the effect of PD173034 through a synergistic or additive mechanism, we performed drug combination analyses by using Chou-Talalay’s algorithm-based isobologram to evaluate dual-agent inhibitory effects and combination indices. As shown by the isobologram, all combination dose-points, except two, were within the right-angled isobologram triangle, indicating olaparib–PD173074 synergism in the PANC-1 cells (Figure 4C). In parallel experiments to elucidate the mechanism underlying the observed anticancer effect of olaparib and/or PD173074, we demonstrated that compared with the mild-to-moderate upregulation of cleaved PARP/PARP, cleaved-caspase (CASP)9/CASP9, and cleaved-CASP3/CASP3 ratios in PANC-1 cells treated with 4.5 μM PD173074 or 3.5 μM olaparib alone, combining PD173074 with olaparib significantly increased the cleaved PARP/PARP, cleaved- CASP9/CASP9, and cleaved-CASP3/CASP3 ratios, concomitantly with a marked decrease in the expression level of Bcl-xL protein (Figure 4D). Thus, indicating, at least in part, that the synergistic effect of PD173074 and olaparib is caspase-dependent and mediated by activation of the CASP3/CASP9/PARP cell death signaling.
Please also kindly see our newly Figure 4 and its legend, page 10, lines 14-25.
Figure 4. Olaparib-induced pharmacological inhibition of PARP1 synergistically enhances the therapeutic effect of PD173074. (A) Graphical representation of changes in PANC-1 and SUIT-2 cell viability when treated with 4.5 μM PD173074 and/or 3.5 μM olaparib for 48 hours. (B) Effect of combining 2.5–10 μM PD173074 and 0.5–5 μM olaparib on the viability of PANC-1 cells. (C) Isobologram and Drug Combination Index Plot indicating synergy between different concentrations of PD173074 and olaparib (D) The effect of treatment with 4.5 μM PD173074 and/or 3.5 μM olaparib on the expression level of PARP, cleaved PARP, pro-caspase 9, cleaved-caspase 9, pro-caspase 3, cleaved-caspase 3, and Bcl-xL proteins as shown by Western blot analysis. *p < 0.05, **p < 0.01, ***p < 0.001 (PANC-1: treated vs CTL); §p < 0.05, §§p < 0.01, §§§p < 0.001 (SUIT-2: treated vs CTL); CTL, control; Comb, PD173074 + Olaparib combination therapy; Fa, fraction affected; CI, combination index; DRI, dose-reduction index.

Round 2
Reviewer 1 Report
The manuscript has been improved in this revised form
Author Response
Point-by-point responses to reviewer’s comments - Reviewer 1:
We would like to thank the reviewer for the thorough reading of our manuscript as well as their valuable comments. We believe all comments are borne out of good faith, and thus, have tried to address their comments conscientiously and feel that they have further improved the readability and appeal of our work, as well as strengthened the manuscript. Below are our point-by-point responses.
Q1. Comments and Suggestions for Authors. The manuscript has been improved in this revised form
A1. We thank the reviewer for taking time to read our manuscript and for the critiques and suggestions made in order to help us improve the quality of our work. In this revised manuscript, we have made effort to make address all the comments and suggestions.

Reviewer 2 Report
Lai et al. have revised extensively to address reviewers’ comments to improve their manuscript. However, there are still concerns and important scientific questions need to be answered or discussed in the manuscript.
To add a mechanistic insight, the authors showed this synergistic effect is caspase-dependent and mediated by activation of the CASP3/CASP9/PARP cell death signaling. However, it sounds a circular reasoning that authors claim that the mechanism of a synergistic effect in cell death is due to activation of cell death signaling. My question was how PARP inhibition can co-operate with FGFR inhibition. Related to this, in pancreatic cancer field, FOLFIRINOX followed by PARP inhibitor (olaparib) as a maintenance therapy becomes a standard of care for BRCA mutant pancreatic cancer patients. It is reasonable to test if FGFR inhibition induces any BRCAness (such as down-regulation of BRA1/2, PALB2 etc) since the synthetic lethal interaction between HR deficiency and PARP inhibition has been shown. If FGFR inhibitor induces HR deficiency, it would be expected. If it is not the case, it requires a new mechanism. Even the authors did not discuss this possibility at all.
The interaction between FGFR1 and PARP1 is interesting, since FGFR1 can be localized in nucleus. However, the physical interaction does not provide any mechanistic insight, although it may provide a clue for future directions. Figure 3D showing the physical interaction by co-IP is still problematic. In Fig3D top panel, IgG control also immunoprecipitates FGFR1, and PARP1 co-IP results are same as IgG IP lane. The same problem applies to the bottom panel. PARP1 IP didn’t seem to work at all, and somehow FGFR1 got differentially immunoprecipitated. Based on the presented data, IP condition is not still optimized and the interaction is not supported. The authors need to provide a convincing evidence that FGFR1 is localized in the nucleus, colocalized with PARP1 (can be shown by immunofluorescence) and physically interact with PARP1 by optimized IP condition. Related to this, the authors need to add insets to show more detailed histology and IHC in Figure 6G. The authors can show a nuclear localization of FGFR1 at least.
Author Response
Point-by-point responses to reviewer’s comments - Reviewer 2:
We would like to thank the reviewer for the thorough reading of our manuscript as well as their valuable comments. We believe all comments are borne out of good faith, and thus, have tried to address their comments conscientiously and feel that they have further improved the readability and appeal of our work, as well as strengthened the manuscript. Below are our point-by-point responses.
Q1. Lai et al. have revised extensively to address reviewers’ comments to improve their manuscript. However, there are still concerns and important scientific questions need to be answered or discussed in the manuscript.
A1. We thank the reviewer for taking time to read our manuscript and for the critiques and suggestions made in order to help us improve the quality of our work. In this revised manuscript, we have made effort to make address all the comments and suggestions.
Q2. To add a mechanistic insight, the authors showed this synergistic effect is caspase-dependent and mediated by activation of the CASP3/CASP9/PARP cell death signaling. However, it sounds a circular reasoning that authors claim that the mechanism of a synergistic effect in cell death is due to activation of cell death signaling. My question was how PARP inhibition can co-operate with FGFR inhibition. Related to this, in pancreatic cancer field, FOLFIRINOX followed by PARP inhibitor (olaparib) as a maintenance therapy becomes a standard of care for BRCA mutant pancreatic cancer patients. It is reasonable to test if FGFR inhibition induces any BRCAness (such as down-regulation of BRA1/2, PALB2 etc) since the synthetic lethal interaction between HR deficiency and PARP inhibition has been shown. If FGFR inhibitor induces HR deficiency, it would be expected. If it is not the case, it requires a new mechanism. Even the authors did not discuss this possibility at all.
A2. We really appreciate the reviewer for the critical insights and this issue is being investigated in our laboratory. We have now included this limitation of our current study in the discussion. We once again, thank the reviewer for providing such a constructive and insightful comment. Please kindly see our revised discussion section, pages 14, lines 41-51 to page 15, line 1-6.
The implication of FGFR1, PARP1, ALDH1A1, ABCB1, RAD51, and H2AFX in therapy response was highlighted by our results. The patients with PDAC were stratified into therapy responders and non-responders. Gene set_high patients exhibiting high FGFR1, PARP1, ALDH1A1, and ABCB1 expression but low RAD51 and H2AFX expression levels were less sensitive to erlotinib and gemcitabine compared to the Gene set_low patients who had low FGFR1, PARP1, ALDH1A1, and ABCB1 expression but high RAD51 and H2AFX expression level (Figure 2). We posit that relative non-responsiveness of the Gene set_high patients may be related to their low mutagenicity, a constitutively active DNA damage response but impaired replication stress response due to probable suppressed levels of phospho-H2AX and RAD51 at baseline [31]; this hypothesis is consistent with our results showing that FGFR1 and PARP1 co-occur and directly interact with each other at the expense of RAD51/H2AFX signaling and with adverse prognostic implications (Figure 3). However, a limitation to this current study remains. It has been established that FOLFIRINOX followed by PARP inhibitor (olaparib) as a maintenance therapy for BRCA mutant pancreatic cancer patients. We have yet to determine if FGFR inhibition could result in the down regulation of BRA1/2 and PALB2, since the synthetic lethal interaction between HR deficiency and PARP inhibition has been shown. This issue is currently being investigated in our laboratory to provide further insights into the mechanistic evidence.
Q3. The interaction between FGFR1 and PARP1 is interesting, since FGFR1 can be localized in nucleus. However, the physical interaction does not provide any mechanistic insight, although it may provide a clue for future directions. Figure 3D showing the physical interaction by co-IP is still problematic. In Fig3D top panel, IgG control also immunoprecipitates FGFR1, and PARP1 co-IP results are same as IgG IP lane. The same problem applies to the bottom panel. PARP1 IP didn’t seem to work at all, and somehow FGFR1 got differentially immunoprecipitated. Based on the presented data, IP condition is not still optimized, and the interaction is not supported. The authors need to provide a convincing evidence that FGFR1 is localized in the nucleus, colocalized with PARP1 (can be shown by immunofluorescence) and physically interact with PARP1 by optimized IP condition.
A3. We thank the reviewer for the encouraging comment and insight. The interaction between FGFR1 and PARP1 is established by immunofluorescence. Please see our newly Figure 3E in our revised manuscript. Please kindly see our revised Results section, pages 8, lines 18-32 to page 9, line 1-9.
3.3. FGFR1 and PARP1 co-occur and directly interact with each other at the expense of RAD51/H2AFX signaling and exhibit adverse prognostic implications
To further characterize the functional association between FGFR1, PARP1, ALDH1A1, ABCB1, RAD51, and H2AFX, as well as to provide a therapeutic context, we performed a co-occurrence and mutual exclusivity analysis using the cBioPortal for cancer genomics platform (https://www.cbioportal.org/). Our results indicate that while FGFR1, PARP1, ALDH1A1, and ABCB1 exhibited varying degrees of co-occurrence, their occurrence was mutually exclusive to that of RAD51 or H2AFX (Figure 3A). Reanalysis of the TCGA-PAAD cohort (n = 176) showed moderate positive correlation between FGFR1 and PARP1 mRNA expression (Spearman’s R: 0.4; Pearson’s R: 0.1) (Figure 3B). Furthermore, using Schrödinger’s PyMOL molecular graphics system (https://pymol.org/2/) for spatiotemporal visualization of probable molecular interaction between FGFR1 and PARP1, we demonstrated that FGFR1 (protein data bank, PDB: 3C4F) directly binds with PARP1 (PDB: 5DS3) with a complementarity score of 18008 to form a stable complex with an interface area of 2482.7Å2 and atomic contact energy (ACE) of 286.24 kcal/mol (Figure 3C). In parallel assays, consistent with the in silico data, results of our co-immunoprecipitation analyses demonstrated direct interaction between FGFR1 and PARP1 in FGFR1-reconstituted PANC-1 and SUIT-2 cells compared with their IgG-treated control counterparts (Figure 3D). By immunofluorescence, we found that PARP1 partially colocalized with FGFR1 in the cell nucleus in the FGFR1-expressing, PARP1-expressing PANC-1 cancer cell line (Figure 3E). In addition, we demonstrated that compared with patients with low PARP1 expression, those with high PARP1 expression exhibited a 1.69-fold worse overall survival (OS) with a mortality index of 0.962 and hazard ratio (HR) = 2.03 (confidence interval, CI: 0.82–5.01) (Figure 3F). Compared to patients with low FGFR1 expression, those with high FGFR1 expression exhibited about 5.25-fold worse OS with a mortality index of 0.974 HR of 8.57 (CI: 2.11–34.91) (Figure 3G).
Please also see our newly Figure 3E and its legend, Page 9, Lines 9-18.
Figure 3. FGFR1 and PARP1 co-occur and directly interact with each other at the expense of RAD51/H2AFX signaling and with adverse prognostic implications. (A) Co-occurrence and mutual exclusivity analysis of FGFR1, PARP1, ALDH1A1, ABCB1, RAD51, and H2AFX in the TCGA Pancreatic Adenocarcinoma cohort. (B) FGFR1 - PARP1 gene expression correlation in the TCGA Pancreatic Adenocarcinoma cohort. (C) Schrödinger’s PyMOL molecular graphics system-generated molecular docking of PARP1 and FGFR1. (D) PANC-1 and SUIT-2 cells were subjected to immunoprecipitation (IP) with FGFR1 (upper) or PARP1 (lower) antibody followed by Western blotting analysis with the indicated antibodies, and IgG serving as control. (E) Interaction of PARP1 with fibroblast growth factor receptor 1 (FGFR1). Coimmunolocalization assays of PARP1 and FGFR1 in the PANC1 cancer cell lines expressing wild-type PARP1 and FGFR1. Kaplan–Meier plots showing the differential overall survival (OS) between (F) FGFR1-low and FGFR1-high samples, or (G) PARP1-low and PARP1-high samples from the TCGA Pancreatic Adenocarcinoma cohort.
Q4. Related to this, the authors need to add insets to show more detailed histology and IHC in Figure 6G. The authors can show a nuclear localization of FGFR1 at least.
A4. We are grateful for the reviewer’s comment. We do believe the reviewer’s concern has now been addressed a nuclear localization of FGFR1 in our newly Figure 6G in our revised manuscript.

This manuscript is a resubmission of an earlier submission. The following is a list of the peer review reports and author responses from that submission.
Round 1
Reviewer 1 Report
I am not able to provide my comments to authors at this moment because of poor quality images was added to the manuscript - the resolution is very low. The legends are too small to be legible. Also, I am not able to evaluate experimental design, Western blot analysis, immunofluorescence etc. I suggest the authors provide images in better resolution to provide my comments.
Reviewer 2 Report
It is very difficult for me to read this paper. The topic is interesting, but the presentation is confused, and the manuscript requires revision.
Reviewer 3 Report
The article entitled "Targeted PARP inhibition in combination with FGFR1 blockade is synthetically lethal to 1 malignant cells in patients with pancreatic cancer" by Lai et al. must be English edited for grammar, style and use since it is very difficult to follow and is not understandable. Furthermore, the manuscript has several flaws in the metodology and conclusions are not supported by results. IC50 doses for in vitro and in vivo testing is not clear, Figures and table are not well distributed, Cut-off points in the Kaplan-Meier curves are not described, etc, etc, etc.
I strongly recommend authors to re-write the whole manuscript. The article must be review carefully by all authors and the English language be edited before a consideration for a futher proffesional peer-review. Therefore, I suggest the article must be rejected.